# Rita Tewari: malaria parasite cell division

Rita Tewari

**Rita Tewari is a Professor of Parasite Cell Biology at The University of Nottingham, Faculty of Medicine and Health Sciences. We asked her about her recent paper published in Life Science Alliance (LSA) and her experience in science thus far.**

Rita Tewari

## 1. How would you explain the main findings of your paper, and how did it come about?

The research in my group is focused on providing new fundamental understanding of malaria parasite cell division, especially during the proliferative sexual stages and transmission within the mosquito vector (Zeeshan et al, 2019a, 2022; Guttery et al, 2022). Although treatment of the human malaria disease is imperative, it is also essential to block malaria parasite transmission to and from mosquitoes to control spread of the infection (Guttery et al, 2022). Unconventional aspects of cell division in the *Plasmodium* sexual stages within the female mosquito gut are a unique feature that is very divergent from model eukaryotes. During its life cycle, the parasite produces centrioles and flagella only in mitosis during male gamete formation (Guttery et al, 2022). These structures are formed de novo during an extremely rapid process in which a haploid gametocyte produces eight flagellated gametes in 15 min, with novel features of cell cycle and microtubule organizing center (MTOC) biology (Sinden et al, 2010; Guttery et al, 2022). Our recent studies in *LSA* have shown that some of the conserved ancestral components of the basal body like SAS4 regulate basal body formation within the cytoplasm of the male cell, although SAS4 is not essential for flagella formation and parasite transmission (Zeeshan et al, 2022). Flagellum formation in this primitive eukaryote is also atypical as it lacks intraflagellar transport (IFT) mechanisms (Sinden et al, 2010).

Using live cell imaging of parasites with dual-tagged fluorescent SAS4/kinetochore marker NDC80 (Zeeshan et al, 2020) and axoneme marker Kinesin 8B (Zeeshan et al, 2019a), we showed that the MTOC in male cells has two components: one cytoplasmic centriolar for axoneme and flagella formation and the other acentriolar for mitotic spindle formation. These components are connected during the rapid mitotic division process leading to male gamete formation, and SAS4 is associated with the centriolar part of the MTOC (Zeeshan et al, 2022).

The SAS4 project evolved because of our curiosity to understand MTOC and flagella assembly during *Plasmodium* male gamete formation. We had shown previously that one of the molecular motor kinesins (kinesin B) plays a role in axoneme biogenesis and is required for axoneme microtubule assembly; however, deletion of the kinesin B gene did not affect spindle formation in the nuclear compartment during cell division (Zeeshan et al, 2019a). To understand the role of the ancestral core proteins of the centriole and basal body in this male cell led to this study.

Most research on the malaria parasite is focused on stages of the life cycle within the vertebrate host and responsible for pathogenesis and disease. However, as we have seen with Covid, pathogen transmission, evolution of drug resistance and the role of global warming are important dimensions in the understanding and development of interventions to control spread of the malaria parasite. Our emphasis is to understand the importance of both evolutionarily conserved and divergent mechanisms of cell division, and to identify potential targets for therapeutic intervention to control parasite proliferation and transmission. For all our work in the laboratory, we collaborate with numerous other groups with expertise in ultrastructure, genome and evolutionary cell biology.

## 2. What was the decision process in choosing where to publish?

The main reasons for choosing this journal was its open access, a rigorous review process, the speed of the process for review and publication, its visibility, and the cooperation of the journal editors, all of which confirmed this as the right place to publish. I was first introduced to *LSA* in 2019 when it was launched by EMBO Press, Rockefeller University Press and Cold Spring Harbor Laboratory Press. The peer review of our earlier publication showed the high standards of the editorial process and referee choices (Zeeshan et al, 2019a). The editors were transparent, understanding, friendly

---

University of Nottingham, Queens Medical Centre, Nottingham, UK

Correspondence: rita.tewari@nottingham.ac.uk

and fair in their judgements, but at the same time robust. For example, the request from the editor for all original images for the figures indicated their rigorous commitment to the process.

### 3. How do you think publishing in an open access journal like Life Science Alliance has impacted the visibility of your findings?

Work on the causative agent of malaria is of global importance. It is important for us to disseminate widely our findings to the scientific community in a way that can be accessed easily and read globally without having to pay for a subscription, particularly in less well-resourced laboratories. It is not only our colleagues in the malaria field, but also scientists from other areas who are interested in cell proliferation and other aspects of life sciences and can access the findings. This accessibility gives huge visibility to the work and removes hurdles to downloading the article such that it can be accessed by many more people, including those working in developing countries where it is difficult or impossible to pay for subscriptions to journals.

### 4. What advice do you have for other researchers on maximizing the dissemination of their work?

I am a signatory of DORA (https://sfdora.org/) and also strongly believe in open access publication. Of course, the alignment of open access journals to preprint servers including BioRxiv (https://www.biorxiv.org/) provides good visibility to the scientific community. Therefore, my advice is to look first for a journal that is best aligned with your work and has robust and constructive editorial and peer review processes. It is also important that the journal is reputable, visible, and respected among peers and judged not just on its impact factor.

Although publication is important, contributing to open science is equally important. There are various ways this can be accomplished, e.g., submitting work to a preprint server, sharing results and ideas at conferences and symposia and discussion with scientists both in and outside the field. This approach gives good exposure and constructive feedback to improve the quality of the data before submission to a journal. There are also social media sites like Twitter, which can also be used in reaching out to scientists globally, and also offer a way to follow the most recent scientific discoveries and knowledge. I also support Reviews Commons which is a very useful platform providing journal-independent peer review and submission to affiliate journals, avoiding multiple rounds of re-review and reformatting. *LSA* is part of this initiative and my experience with this platform has been very useful.

### 5. What questions is your lab currently trying to answer?

My lab focuses on understanding the fundamental biology of cell division in proliferation and transmission of the malaria parasite, as well as the role of cell polarity in parasite movement and invasion of host cells and tissues. Presently, we are studying mitosis in male gamete formation from a gametocyte that features very rapid cell division. Genome replication from 1N to 8N occurs in 8 min, with successive mitotic spindle and axoneme formation, nuclear

division and exflagellation of the atypical, flagellated gametes completed in 12–15 min. This rapidity suggests that the cell cycle, MTOC and centriole biology have some unique features compared to model organisms. What regulates and governs these processes is our major interest. My group performed the first protein kinome and phosphatome screens revealing the presence of divergent kinases and phosphatases, as well the absence of some of the classical controllers of mitosis like polo kinases, CDC25 and CDC14 from the malaria genome (Tewari et al, 2010; Guttery et al, 2014). In addition, we showed that Plasmodium has a small complement of cyclins and a streamlined anaphase promoting complex (Guttery et al, 2012; Roques et al, 2015; Wall et al, 2018).

These studies have provided the resources to unravel the unique biology of cell division in the malaria parasite. I am very pleased to have been recently awarded an ERC advanced grant, allowing me to undertake this research and combine cutting-edge cell biology and collaborations with evolutionary scientists, structural biologists and experts on spindle biology in other eukaryotic systems. These studies and collaborations will be key in obtaining a holistic view of this unique cell.

In addition, we are interested in another stage of parasite development where meiosis occurs (the ookinete). Following fertilization, the zygote differentiates to form the ookinete, a motile invasive stage within the mosquito gut (Guttery et al, 2022). We have shown previously that this cell has a distinct apical-basal polarity by studying certain pellicle components and molecular motors like myosins and kinesins (Poulin et al, 2013; Saini et al, 2017; Wall et al, 2019; Zeeshan et al, 2019b). The development and regulation of polarity in this cell and how this process proceeds simultaneously with meiosis in ookinete development is a fascinating area to study and understand, and forms the basis of another project that is currently being undertaken in our group.

### 6. What motivated you to pursue a career in science, and what have been the most interesting moments on the path that led you to where you are now?

I grew up in India where I did my graduate studies. My grandmothers had never gone to school and could neither read nor write. However, they were very analytical, liberal, and open in terms of their views and living. My mother believed that education and science can make a more equal society, and her motivation was so inspiring that both my sister and I pursued a career in science. My father was a professor in social sciences and so was embedded in academic life. I was inspired by the intricacies of cell biology, particularly in relation to human disease, and this fascination was the guiding force for my career path. For my PhD, I was given a rigorous training by my supervisor, Prof SRV Rao at the University of Delhi. The research in his lab was my formative years in cytogenetics and cell biology to understand fragile sites in chromosome biology. During my graduate studies, I was encouraged to visit various European labs, to present my work, and to learn cutting-edge approaches in labs outside of India (e.g., in MPI Freiburg at Germany, IMP at Austria and Nagoya University Japan). I was very fortunate to have Prof Rao as my first mentor. My research career continued in cell biology and molecular biology with a first post-doctoral fellowship at INRA France and then a

second one mentored by Prof Frank Grosveld in Rotterdam, Holland, where I learned hardcore molecular and cell biology approaches to understand β-globin gene regulation in red blood cell biology. These two periods of research training in India and in Holland shaped me as a cell and developmental biologist. Then, in my own independent research career, I took on the challenge to understand the cell biology of malaria parasite development within the mosquito, a niche I first occupied at Imperial College London, UK.

My most important and defining moment was at a crossroads I experienced following my post-doctoral career after a short stay in Bergen, Norway and before taking up the research lecturer position to study malaria biology in London. Although I had experienced malaria disease as a child many times, I had very little knowledge about the parasite's biology. Hence, to embark on a career in parasite cell biology was a big challenge, but it was also a question of survival for my research career. At Imperial College with Dr. Oliver Billker, I carried out a most challenging functional screen of the entire malaria parasite protein kinome, and this experience gave me the understanding and knowhow to enable me to develop my group at the University of Nottingham. Since 2009, my group has been pursuing many research projects to understand the regulation and dynamics of Plasmodium cell division and cell polarity in driving parasite proliferation and invasion, and thereby transmission. During my independent career, I have had an excellent mentorship and collaboration with Dr. Tony Holder at the National Institute for Medical Research, Mill Hill and The Crick Institute, who works on human malaria parasite species.

### 7. Tell us something interesting about yourself that wouldn't be on your CV

I am a very sociable person and believe in the value of lots of interactions and learning the cultures, languages, and culinary dishes of different countries. Although I grew up in India, for the last three decades I have resided in Europe. Although I face many challenges, sometimes also very racist and discriminatory treatments, I always believed that my openness both as a person and in science with my work and curiosity is my refuge and strength. I have imbibed the knowledge and learned how to make many culinary dishes from these countries, and along the way, I have made many long-lasting friends which is my real wealth. I strongly believe in United Europe and feel science and scientist do not have boundaries in their pursuit for knowledge. Hence, now my cooking has the spices and fusion of India and Europe. I can understand and speak some European languages like French, German, Norwegian and Dutch, as well as Indian languages like Hindi, Punjabi, Bengali and of course English (both Indian and UK varieties - just joking!).

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
