## [Reviewer comments · Life Science Alliance]

Life Science Alliance

Rita Tewari: Malaria Parasite Cell Division

Rita Tewari

DOI: <https://doi.org/10.26508/lsa.202201682>

Corresponding author(s): Rita Tewari, University of Nottingham

Review Timeline:

Submission Date:

2022-08-19

Accepted:

2022-08-19

Transaction Report:

August 19, 2022

RE: Life Science Alliance Manuscript #LSA-2022-01682

Prof. Rita Tewari
University of Nottingham
School of Life Sciences
Queens Medical Centre
Nottingham NG7 2UH
United Kingdom

Dear Dr. Tewari,

Thank you for submitting your Interviews entitled "Rita Tewari: Malaria Parasite Cell Division". It is a pleasure to let you know that your manuscript is now accepted for publication in Life Science Alliance.

Your manuscript will now progress through copyediting and proofing.

Again, thank you for participating in this project.

Sincerely,

August 19, 2022

RE: Life Science Alliance Manuscript #LSA-2022-01682

Prof. Rita Tewari
University of Nottingham
School of Life Sciences
Queens Medical Centre
Nottingham NG7 2UH
United Kingdom

Dear Dr. Tewari,

Thank you for submitting your Interviews entitled "Rita Tewari: Malaria Parasite Cell Division". It is a pleasure to let you know that your manuscript is now accepted for publication in Life Science Alliance.

Your manuscript will now progress through copyediting and proofing.

Again, thank you for participating in this project.

Sincerely,
